# An Inducible Luminescent System to Explore Parkinson’s Disease-Associated Genes

**DOI:** 10.3390/ijms25179493

**Published:** 2024-08-31

**Authors:** Anelya Gandy, Gilles Maussion, Sara Al-Habyan, Michael Nicouleau, Zhipeng You, Carol X.-Q. Chen, Narges Abdian, Nathalia Aprahamian, Andrea I. Krahn, Louise Larocque, Thomas M. Durcan, Eric Deneault

**Affiliations:** 1The Neuro’s Early Drug Discovery Unit (EDDU), McGill University, 3801 University Street, Montreal, QC H3A 2B4, Canada; anelya.gandy@mail.mcgill.ca (A.G.); gilles.maussion@mcgill.ca (G.M.); michael.nicouleau@mcgill.ca (M.N.); zhipeng.you@mcgill.ca (Z.Y.); xiuqing.chen@mcgill.ca (C.X.-Q.C.); narges.abdian@mcgill.ca (N.A.); nathalia.aprahamian@mcgill.ca (N.A.); andrea.krahnroldan@mcgill.ca (A.I.K.); 2Centre for Oncology, Radiopharmaceuticals and Research (CORR), Biologic and Radiopharmaceutical Drugs Directorate (BRDD), Health Products and Food Branch (HPFB), Health Canada, Ottawa, ON K1A 0K9, Canadalouise.larocque@hc-sc.gc.ca (L.L.)

**Keywords:** HiBiT-LgBiT technology, cell-based assays, iPSC-derived models, *GBA1*, Parkinson’s disease

## Abstract

With emerging genetic association studies, new genes and pathways are revealed as causative factors in the development of Parkinson’s disease (PD). However, many of these PD genes are poorly characterized in terms of their function, subcellular localization, and interaction with other components in cellular pathways. This represents a major obstacle towards a better understanding of the molecular causes of PD, with deeper molecular studies often hindered by a lack of high-quality, validated antibodies for detecting the corresponding proteins of interest. In this study, we leveraged the nanoluciferase-derived LgBiT-HiBiT system by generating a cohort of tagged PD genes in both induced pluripotent stem cells (iPSCs) and iPSC-derived neuronal cells. To promote luminescence signals within cells, a master iPSC line was generated, in which LgBiT expression is under the control of a doxycycline-inducible promoter. LgBiT could bind to HiBiT when present either alone or when tagged onto different PD-associated proteins encoded by the genes *GBA1*, *GPNMB*, *LRRK2*, *PINK1*, *PRKN*, *SNCA*, *VPS13C*, and *VPS35*. Several HiBiT-tagged proteins could already generate luminescence in iPSCs in response to the doxycycline induction of LgBiT, with the enzyme glucosylceramidase beta 1 (GCase), encoded by *GBA1*, being one such example. Moreover, the GCase chaperone ambroxol elicited an increase in the luminescence signal in HiBiT-tagged *GBA1* cells, correlating with an increase in the levels of GCase in dopaminergic cells. Taken together, we have developed and validated a Doxycycline-inducible luminescence system to serve as a sensitive assay for the quantification, localization, and activity of HiBiT-tagged PD-associated proteins with reliable sensitivity and efficiency.

## 1. Introduction

PD represents the second most common neurodegenerative disease, affecting >2% of the population above the age of 65 [1]. The discovery of 90 risk variants across 78 genomic loci that can cause or confer the risk of developing PD has provided important insight into the neurobiology of the disease [2]. A growing number of genes are involved in these associations, and understanding their underlying mechanisms will help in finding treatments. PD pathology is characterized by the presence of cytoplasmic inclusions within neurons called Lewy bodies, formed by the misfolding of the protein α-syn, encoded by *SNCA*, the first PD-associated gene to be discovered >25 years ago [3,4]. Aggregated α-syn can spread from one neuron to another in a prion-like fashion [5]. Significant progress has been made towards understanding the function of additional genes, including but not limited to *PRKN*, *PINK1, GBA1*, and *LRRK2* [6,7,8]. Strikingly, only a handful of known PD genes are truly characterized. Understanding the role, localization, and interaction of both the well-known and many of the lesser-studied PD genes is imperative to better understand cellular pathways underlying the development of PD.

A significant hurdle in studying the function and expression of these PD-associated proteins is often a lack of specific and validated research-grade antibodies (Figure 1A). This limits our understanding of where these PD-associated proteins are localized, how they are expressed across cells and conditions, and whether we can develop therapies to regulate their activity [9,10,11]. Moreover, the detection of a number of PD proteins involved in cellular signaling is often challenging due to low expression levels across brain cells [12] (Figure 1A). To overcome these issues, endogenous protein expression, localization, modification, and interaction can now be characterized at physiological levels through reporter gene fusions using emerging gene editing techniques such as the clustered regularly interspaced short palindromic repeat (CRISPR) system [13]. Gene editing of endogenous genes means we can now tag specific proteins, circumventing potential issues from their overexpression [14] (Figure 1A). The large size of fluorescent reporter genes, such as green fluorescent protein (GFP), often interferes with the normal function of a protein [15] (Figure 1A), meaning that smaller tags are favored to reduce the potential impact on the function of the tagged protein. Epitope tags, including FLAG, HA, or Myc, are more easily and rapidly inserted using CRISPR due to their small size [16], a difference of ~10 amino acids vs. ~240 amino acids for GFP, that can be included within a DNA donor template synthesized as a single-stranded oligodeoxynucleotide (ssODN). Nonetheless, such epitope tags normally require cell fixation or cell lysis and specific antibodies, allowing only endpoint analysis and precluding real-time analysis of the endogenous protein in living cells.

For real-time monitoring of localization and expression levels of a protein of interest in living cells, split luciferase-based reporters have evolved to provide more sensitive detection and faster quantitation over an extended concentration range [17]. For instance, HiBiT is a small 1.3 kDa subunit binding with high affinity to a larger 18 kDa subunit, named LgBiT, to restore the function of the luciferase enzyme NanoLuc, which emits luminescence in the presence of its substrate [18,19]. LgBiT protein may be added manually to the cell lysates in a lytic detection approach, or it can be expressed exogenously for live imaging purposes. The small HiBiT fragment is well-suited for the CRISPR tagging of proteins with diverse functions and subcellular localizations across various cell types. As a demonstration of this technology, several proteins involved in different signaling pathways were successfully tagged with HiBiT for diverse expression characterization studies across multiple conditions [19].

In this study, we hypothesized that HiBiT fusions to endogenous PD risk genes can generate new cellular models to provide novel insights into the expression and function of a number of PD-associated proteins. We leveraged the CRISPR gene editing system to fuse the HiBiT sequence at the 5′ or 3′ end of the coding sequence for a series of PD candidate genes in human induced pluripotent stem cells (iPSCs). A luminescence signal revealed the expression of the tagged proteins. The choice of target genes depended on their relevance to PD, availability of reliable antibodies for detecting the endogenous gene product, and pertinence to live cell imaging and protein quantitation studies. Validated iPSCs, in which the tags have been correctly inserted, were used to derive dopaminergic neural progenitor cells (dNPCs) and dopaminergic neurons (DNs), providing human disease-relevant cell types for studying the real-time expression and localization of these PD-associated proteins, with a focus on *GBA1* in this study. Thus, the work represents an essential step towards understanding the key molecular mechanisms by which proteins may be implicated in the development of PD.

## 2. Results

### 2.1. Engineering a Master Doxycycline-Inducible LgBiT iPSC Line

Generating representative cellular models with the LgBiT-HiBiT reporter assay system to accurately quantify protein dynamics requires the presence of both nanoluciferase subunits, i.e., HiBiT and LgBiT, in proximity to emit luminescence when its substrate is present (Figure 1B). The expression of the small HiBiT subunit is dependent on the expression of the target endogenous gene to which it is appended. However, the LgBiT subunit needs to be added manually to the cell lysates when using a lytic detection approach or expressed via exogenous nucleic acids for live imaging purposes. Since neurons are typically difficult to transfect, and lentiviral vectors risk inducing insertional mutagenesis, we opted to generate a monoclonal human iPSC line carrying a LgBiT expression cassette in a safe-harbor locus for real-time and endpoint studies of our tagged proteins. For this, we engineered our well-characterized control iPSC line AIW002-02 [20]. We selected the safe-harbor locus Citramalyl-CoA Lyase (*CLYBL*) for its persistence of activity upon neuronal differentiation, compared with other safe-harbor loci, such as *AAVS1* [21]. We used the CRISPR-Cas9 nuclease and a plasmid template to insert the LgBiT DNA sequence into *CLYBL*. Of note, we constructed a LgBiT sequence made of codons specifically optimized for human cells, along with a Kozak sequence upstream, for increased expression. Moreover, a Doxycycline (Dox)-inducible promoter was used to drive the tightly controlled expression of LgBiT (hereafter named iLgBiT) at different steps of cell differentiation (Figure 1C).

### 2.2. iLgBiT Copy Number Assessment and Clonal Validation

Following transfection of the CRISPR machinery into AIW002-02 iPSCs, three weeks of Neomycin treatment to eliminate non-edited cells, and Cre recombinase treatment to remove the Neomycin selection cassette, several single cell-derived iPSC clones were isolated using flow cytometry and screened for the correct insertion of the iLgBiT cassette into the *CLYBL* locus. We used droplet digital polymerase chain reaction (ddPCR) to assess the copy numbers of inserted iLgBiT sequence per genome into the newly isolated Neomycin-resistant clones, compared with a control autosomal endogenous gene, i.e., *SYT1* (Figure 1D). We reasoned that an allele ratio iLgBiT:SYT1 of 0.5 would correspond to one iLgBiT allele per two SYT1 alleles, representative of a heterozygous iLgBiT-CLYBL cell population. In contrast, an allele ratio of 1.0 would indicate a homozygous iLgBiT-CLYBL population. Furthermore, a ratio of >1.0 would imply the presence of cells carrying 3 or more iLgBiT cassettes, i.e., at least one random off-target insertion. For example, clone M1G4 presented an allele ratio iLgBiT:SYT1 of 0.89 (Figure 1E); thus, we speculated a mix of cells carrying different numbers of the iLgBiT allele, which likely required at least one additional round of flow cytometry purification to reach monoclonal purity. Likewise, we calculated the allele ratio iLgBiT:SYT1 for the 21 other clones that emerged as single colonies in separate wells, as observed under the microscope. These clones presented a range of ratios from 0.08 to 1.69 (Figure 1E), suggestive of various iLgBiT copy numbers among the different cell populations. From these ddPCR results, we selected clones M1C6 and M2E3 with a ratio close to 0.5 (single copy of iLgBiT), as well as M1D5 and M1G4 close to 1.0 (two copies), for further validation experiments (highlighted in yellow in Figure 1E).

To validate the clonal constitution of the selected clones, we used flow cytometry to sort single cell-derived iPSC subclones from each selected cell population in a second round of purification. The allele ratio iLgBiT:SYT1 was calculated for all newly emerged subclones. Surprisingly, the average ratio iLgBiT:SYT1 from the 14 different subclones derived from M1C6 was 3.0 (Table 1). Unfortunately, this implies the presence of six iLgBiT alleles per genome, including four iLgBiT copies inserted randomly at off-target locations. This potentially involved insertional mutagenesis. Since the ratio was 0.41 for M1C6 during the first round of purification (Figure 1E), we suspect that a six-copy and hyperproliferative subclone took over cell culture during the second round.

By contrast, we derived 16 different subclones from M2E3, and all carried a single insertion of iLgBiT, as supported by an average allele ratio iLgBiT:SYT1 of 0.5, with the lowest value at 0.4 and the highest at 0.7 (Table 1). This indicates that clone M2E3 is both monoclonal and heterozygous.

From the 16 subclones derived from M1G4, we could detect the presence of at least two distinct clones. Indeed, 13 subclones presented a ratio of 0.4–0.5, while the other three subclones were between 2.5 and 2.8 (Table 1). This strongly suggests that most (81%) of the M1G4 cell population is carrying a single iLgBiT copy per genome (ratio 0.5), while the rest is carrying at least five copies (ratio 2.5). Thus, M1G4 cells would require an additional purification step to eliminate the five copy cells (potentially abnormal due to insertional mutagenesis) and maintain only the one-copy cell fraction.

### 2.3. Confirmation of Heterozygosity

To ensure that at least one copy of iLgBiT was properly inserted on-target into *CLYBL*, we designed a set of PCR primers close to the *CLYBL* insertion site, located upstream of the left homology arm and within the Neomycin resistance cassette (see primers CLY5-F2 and Neo-R2 in Figure 1F). The resulting amplified fragment was expected at 1095 bp only in the presence of the Neomycin selection cassette, i.e., without proper Cre recombination. Clones M1D5 and M1C6 were found to be positive by PCR for the presence of the Neomycin cassette at the *CLYBL* insertion site (Figure 1F), indicating that the Cre excision was not complete on at least one allele. By contrast, the Neomycin cassette was completely removed in clones M1G4, M2E3, and M2A3, as assessed by the absence of PCR amplification (Figure 1F).

We designed another primer set to amplify a 1111 bp PCR fragment to detect the appropriate excision of the Neomycin cassette out of the *CLYBL* insertion site, following Cre recombinase treatment (see primers CLY5-F1 and Cre-R1 in Figure 1F). All tested clones, except M1D5, presented with the expected band, supporting the elimination of the Neomycin cassette from at least one allele (Figure 1F).

In parallel, we devised an additional set of primers to generate a 1813 bp PCR fragment corresponding to the wt allele, i.e., without any insert into the *CLYBL* insertion site (see primers CLYwt-F2 and CLYwt-R2 in Figure 1F). All tested clones were positive for the wt allele (Figure 1F), implying that at least one *CLYBL* allele was kept away from iLgBiT insertion. These results support the heterozygosity of iLgBiT insertion into *CLYBL*, as well as the complete removal of the Neomycin cassette for M1G4 and M2E3.

### 2.4. Expression of Active iLgBiT upon Doxycycline Treatment

With the master M2E3 iPSC now generated, we next performed a series of quality control steps to ensure the cell line was behaving like a normal iPSC with normal genome stability. As a first step, the M2E3 iPSCs were demonstrated to display uniform colony formation (Appendix A) and had a normal karyotype with no detectable genomic instabilities (Figure 2A,B). Immunocytochemistry (ICC) revealed positive staining for pluripotency markers Nanog, Tra-160, SSEA-4, and OCT3/4 in M2E3 iPSCs (Figure 2C). RT-qPCR analysis corroborated the expression of Nanog (*t*-test score [t] = 11.47, degrees of freedom [df] = 4, * *p*-value [p] < 0.002) and OCT3/4 (t = 24.21, df = 4, * *p* < 0.002) in M2E3 iPSCs (Figure 2D). Following neural induction into progenitors (Appendix A), M2E3 dNPCs stained positive for SOX1, FOXA2, and Nestin (Figure 2E). RT-qPCR analysis confirmed the expression of progenitor and dopaminergic markers Nestin (t = 40.36, df = 4, * *p* < 0.002), FOXA2 (t = 21.18, df = 4, * *p* < 0.002), EN1 (t = 8.978, df = 4, * *p* < 0.002), TH (t = 27.76, df = 4, * *p* < 0.002), and LMX1 (t = 7.475, df = 4, * *p* < 0.002) in M2E3 dNPCs (Figure 2D).

We also measured the baseline mRNA expression of *GBA1* in the M2E3 line (Appendix A). As expected with differentiation from iPSCs to dNPCs, there was a significant increase in *GBA1* expression across cell types (t = 12.78, df = 4, * *p* < 0.002).

Next, we validated the ability of M2E3 iPSCs and dNPCs to increase iLgBiT expression upon Dox treatment using ICC and RT-qPCR. ICC showed that Dox treatment increased iLgBiT expression in M2E3 iPSCs (Figure 2F) and dNPCs (Figure 2G). RT-qPCR analysis confirmed a significant increase in Dox-induced iLgBiT expression in iPSCs (ANOVA score (F) = 84.57, df = 1.8, * *p* < 0.002), as well as a modest increase in dNPCs (Figure 2H). Furthermore, the presence of iLgBiT was clearly revealed by a significant increase in luminescence in M2E3 dNPCs treated with Dox and varying concentrations of HiBiT control protein (Figure 2I). The addition of LgBiT control protein, without any added HiBiT or Dox, increased the RLU from 9 to 421 (Figure 2I). This suggests that the added LgBiT may present some residual luciferase activity on its own, i.e., without any HiBiT complementation, especially at high concentrations. The addition of Dox, on top of LgBiT control protein, did not increase the signal further in M2E3 NPCs (Figure 2I), suggesting a saturation of LgBiT residual activity (when added at high concentration) that may mask the detection of low HiBiT levels, and decrease the sensitivity of this approach.

### 2.5. HiBiT Tagging to PD Genes into Master M2E3 iPSC Line

Flanking HiBiT to any target gene necessitates the insertion of its 33 bp DNA fragment into genomic DNA. To achieve this in our master M2E3 iPSC line, we used the CRISPR-Cas9 system coupled with HDR, along with sgRNAs and ssODN templates specifically designed for each target gene. We selected eight target genes that were previously associated with PD, namely *GBA1*, *GPNMB*, *LRRK2*, *PINK1*, *PRKN*, *SNCA*, *VPS13C*, and *VPS35*, and for which live cell imaging of the corresponding endogenous protein may represent a challenge using currently available antibodies [22,23]. The HiBiT sequence was fused onto the N- or C-terminal of each translated protein, depending on available information in the literature about possible interference with the native protein function. As examples, the N-terminal domain of α-syn is responsible for interaction with substrate proteins, while the C-terminal domain of α-syn may be involved in the inhibition of its aggregation and degradation [24]. Parkin can perform autoubiquitination and can be maintained in an autoinhibited state [25], which was shown to be disrupted in the presence of small epitope tags like cMyc, FLAG, or HA (all similar in size to HiBiT) fused to the N-terminus of Parkin [26]. GPNMB is a transmembrane glycoprotein with a cytoplasmic C-terminal tail [27]. LRRK2 has N-terminal domains that play crucial roles in its function [28]. Interactions between the N- and C-terminus of PINK1 are critical for its stabilization at the outer mitochondrial membrane [29]. Given the well-characterized roles α-syn and Parkin have in PD, we decided to make both an N- and a C-terminal fusion of HiBiT for each. All other HiBiT fusions were made only in the C-terminus to save time and resources and to limit the likelihood of function interference.

In total, 10 HiBiT cell-line fusions were made, and we named each fusion according to the location of HiBiT relative to the target gene. For instance, a HiBiT positioned in front of the target gene name, e.g., HiBiT-SNCA, means that the HiBiT sequence was fused to the N-terminal of its corresponding protein, while HiBiT following the gene name, e.g., SNCA1-HiBiT, refers to a C-terminal fusion. Following the nucleofection of the CRISPR components for HiBiT insertion into our master M2E3 iPSC line, several iPSC clones were isolated and screened for the correct HiBiT tagging onto each target gene (Figure 3A). We used ddPCR to monitor the copy number of the HiBiT sequence inserted per genome in each isolated iPSC clone by comparing it to the control autosomal endogenous *SYT1* (Figure 3B). For each target gene, we aimed to isolate one monoclonal iPSC clone presenting with similar copy numbers of HiBiT and SYT1, i.e., carrying two copies of HiBiT. Eight isolated clones out of ten presented two HiBiT alleles and two SYT1 alleles, resulting in an allele ratio of HiBiT:SYT1 around 1.0, i.e., GPNMB-HiBiT, LRRK2-HiBiT, HiBiT-PRKN, PRKN-HiBiT, HiBiT-SNCA, SNCA-HiBiT, VPS13C-HiBiT, and VPS35-HiBiT (Figure 3C). However, we detected an allele ratio of HiBiT:SYT1 of 2.0 (four HiBiT alleles) in GBA1-HiBiT (Figure 3C), probably due to the concomitant insertion of HiBiT into the pseudogene *GBA1LP*, which presents 100% homology with *GBA1* in the targeted area. PINK1-HiBiT was the other exception, where the corresponding cells presented an allele ratio HiBiT:SYT1 less than 0.5 (Figure 3C), suggesting that PINK-HiBiT was heterozygous and probably not monoclonal yet.

To confirm an error-free insertion of the intact HiBiT DNA sequence, each targeted locus was amplified by PCR and sequenced (Figure 3D). Each N-terminal fusion was correctly engineered with HiBiT inserted right after the start codon (see green frame in Figure 3D), while each C-terminal fusion was encoded by HiBiT placed just before the stop codon (see red frame in Figure 3D). All presented a perfect assemblage, except for PINK1-HiBiT, which was aberrant with at least two different alleles, i.e., one HiBiT allele, and the other(s) either wt or affected by an on-target small insertion/deletion (indel) surrounding the cut site (Figure 3D).

We evaluated the ability to produce luminescence upon Dox induction in the 10 newly generated HiBiT fusions in iPSCs. As shown in Figure 3E, five lines produced a significant increase in luminescence in the presence of Dox, i.e., GBA1-HiBiT, GPNMB-HiBiT, HiBiT-SNCA, SNCA-HiBiT, and VPS13C-HiBiT. We supposed that baseline expression of the other target genes is not high enough in iPSCs, at least in our conditions, to make a difference in luminescence, e.g., *LRRK2* and *PRKN*, and/or that HiBiT is less accessible for LgBiT in these cases. We also tested the effects of adding an unlimited amount of LgBiT control protein, with or without Dox treatment, to see if iLgBiT was limiting the extent of luminescence production by the HiBiT-tagged proteins. As expected, the luminescence signal was increased compared to Dox alone, at least for GBA1-HiBiT, HiBiT-SNCA, SNCA-HiBiT, and VPS13C-HiBiT (Appendix A). This suggests that the amount of iLgBiT produced in our Dox-inducible system was limiting the extent of luminescence that could be produced by these three highly expressed proteins in iPSCs. However, in the case of GPNMB-HiBiT, the addition of LgBiT control protein did not increase the signal much further (Appendix A), suggesting that the amount of Dox-induced iLgBiT was not limiting here, probably because GPNMB-HiBiT was not highly expressed. More importantly, our control M2E3 (no HiBiT), with or without Dox, showed an increase in RLU from about 10 to 1000 following the addition of LgBiT control protein (Figure 3E and Appendix A). The same observation was made for all the other conditions that were considered baseline, i.e., around 10 or less (Figure 3E and Appendix A). As explained above, this is likely due to the known residual activity of LgBiT on its own, i.e., without any HiBiT complementation, especially at high concentrations of added LgBiT control protein. Therefore, we reasoned that the high levels of residual activity of added LgBiT were actually masking the detection of GPNMB-HiBiT luminescence after Dox treatment (Appendix A). In other words, GPNMB-HiBiT would represent a false negative in the presence of added LgBiT. We think that the addition of LgBiT was likely restricting the sensitivity of our system in iPSCs.

We decided to pursue the following characterization experiments with the GBA1-HiBiT line only for the following reasons: (i) these cells presented a significant increase in luminescence (Figure 3E); (ii) the function of *GBA1* is already well known; (iii) drugs are commercially available and known to increase GCase activity, which improves lysosomal function and prevents α-syn aggregation; (iv) *GBA1* represents the most common genetic risk factor for PD. 

### 2.6. Characterization of GBA1-HiBiT iPSCs, dNPCs, and DNs

GBA1-HiBiT iPSCs displayed the expected iPSC morphology (Appendix A) and normal karyotyping with no detectable genomic instabilities (Figure 4A,B). ICC revealed positive staining for pluripotency markers Nanog, Tra-160, SSEA-4, and OCT3/4 in GBA1-HiBiT iPSCs (Figure 4C). All other HiBiT-tagged iPSC lines also passed these quality control tests. We confirmed a significant expression in the pluripotency markers Nanog (t = 43.36, df = 4, * *p* < 0.005) and OCT3/4 (t = 11.77, df = 4, * *p* < 0.005) in GBA1-HiBiT iPSCs, compared with dNPCs by RT-qPCR (Figure 4D). GBA1-HiBiT iPSC-derived dNPCs stained positive by ICC for SOX1, FOXA2, and Nestin (Appendix A and Figure 4E). RT-qPCR further confirmed a significant increase in the expression of Nestin (t = 11.36, df = 4, * *p* < 0.005), FOXA2 (t = 13.40, df = 4, * *p* < 0.005), EN1 (t = 30.60, df = 4, * *p* < 0.005), and TH (t = 5.616, df = 4, * *p* < 0.005) upon differentiation of GBA1-HiBiT iPSCs into dNPCs (Figure 4D). Next, we validated the Dox-inducible iLgBiT expression in GBA1-HiBiT cells. After Dox treatment, ICC revealed positive staining for iLgBiT in GBA1-HiBiT iPSCs (Figure 4F) and in dNPCs (Figure 4G). RT-qPCR confirmed a significant increase in iLgBiT mRNA expression after Dox treatment (F = 186.1, df = 1.8, * *p* < 0.005) in both GBA1-HiBiT iPSCs and dNPCs (Figure 4H).

We also measured the expression of *GBA1* mRNA transcripts in the GBA1-HiBiT line. RT-qPCR results showed no difference in *GBA1* expression after Dox treatment in both iPSCs and dNPCs (Figure 4I), indicating that Dox treatment does not directly interfere with *GBA1* expression (F = 0.2414, df = 1.7, *p* = 0.638). Moreover, we detected a significant increase in the expression of *GBA1* in dNPCs compared with iPSCs (Figure 4I [F = 10.12, df = 1.7, * *p* < 0.05], and Appendix A), which was expected in normal conditions. Immunoblotting further supported the expression of Dox-inducible iLgBiT, HiBiT, and GCase in GBA1-HiBiT iPSCs and dNPCs (Figure 4J).

### 2.7. Validation of iLgBiT-HiBiT System with Pharmacological Chaperone Ambroxol

As specific molecules are available to enhance GCase levels and activity, we wanted to evaluate whether the impact of such a molecule could be detected with this luminescent system. Ambroxol, a mucolytic agent for the treatment of respiratory diseases, has been shown to function as a pharmacological chaperone of the GCase enzyme, enhancing its levels and activity at the mRNA and protein levels [30]. We sought to determine whether Ambroxol can be used as a positive control for functional validation of our iLgBiT-HiBiT luminescence system. As expected, we observed a significant increase in luminescence in GBA1-HiBiT dNPCs treated with 70 µM Ambroxol for 6 days, compared with those treated with DMSO (F = 401.3, df = 2.3, * *p* < 0.03), while the master M2E3 line, lacking HiBiT, produced negligible luminescence (Figure 5A). Immunoblotting only detected a modest increase in HiBiT expression in GBA1-HiBiT dNPCs after Ambroxol treatment (Appendix A). This potentially highlights a greater sensitivity of the luminescence system for detecting slight increases that may be undetectable on immunoblot.

Furthermore, we differentiated M2E3 and GBA1-HiBiT dNPCs into DNs (Appendix A). We observed a significant increase in luminescence following Ambroxol treatment in GBA1-HiBiT DNs, compared with DMSO-treated DNs (F = 1829, df = 2.3, * *p* < 0.03), while there was negligible luminescence accounted for by M2E3 DNs (Figure 5B).

In support of this increased luminescence in DNs in the presence of Ambroxol, we used a fluorescent 4-methylumbelliferyl-β-D-galactopyranoside (4-MUG) kinetic assay that monitors galactosidase activity and showed a significant increase in GCase enzyme activity in DNs after Ambroxol treatment (t = 5.649, df = 2, * *p* < 0.03) (Figure 5C). These results support the notion that GCase enzyme activity is enhanced by Ambroxol treatment in DNs.

### 2.8. Off-Target Activity Analysis of CLYBL and GBA1 sgRNAs

One of the main limitations of gene editing using the CRISPR-Cas9 nuclease system is the potential of off-target DSB-induced indel activity. If a genomic region is similar to a sgRNA sequence, with an adjacent PAM site, Cas9 targeting of that region is possible depending on the number of mismatches present compared with the on-target site. We used benchling.com to locate the top three candidate genomic sites that could have most likely been targeted, and then PCR amplified each region surrounding the potential cut sites with expected band sizes (Appendix A). Sanger sequencing of the six different regions confirmed full alignment with the wild-type counterparts without any mutations detected (Appendix A). These results suggest that no indels were present at the three most likely sites for off-target activity for both *CLYBL* and *GBA1* sgRNAs and that these sites maintained their integrity during the editing process.

## 3. Discussion

In this study, we have engineered a human iPSC line to express the nanoluciferase subunit LgBiT in a tunable fashion to complement selected target proteins that were tagged with the small subunit HiBiT. The goal was to characterize the expression, activity, and subcellular localization of the nascent luminescent proteins, and our findings with *GBA1* support the use of this system for monitoring the expression of a tagged protein. We showed that both luminescence and GCase enzyme activity are enhanced by Ambroxol treatment in DNs, which provides support for the use of Ambroxol as a positive control in candidate screens, using luminescence as a readout to identify regulators of GCase activity in PD. For example, a recent preprint explored the relevance of the HiBiT system in a high-content screen to identify new drugs that enhance GCase [31].

The choice to insert the inducible LgBiT cassette into the safe-harbor site *CLYBL* was motivated by two reasons. First, exogenous DNA insertions such as donor templates or viral vector integrations can be recognized as structural variants at the origin of insertional mutagenesis driving tumorigenesis [32], which has the potential to corrupt any experiment. However, the insertion of large cassettes into safe-harbor loci was shown to prevent such catastrophic events [33] and should represent a key consideration for any gene therapy. For this reason, we aimed to isolate iPSC clones carrying a single LgBiT insertion precisely into the safe-harbor site *CLYBL*. Second, the *CLYBL* safe-harbor locus was selected for its superior persistence of transgene expression throughout neuronal differentiation, as compared with other loci such as AAVS1 [21]. Although Dox was able to induce a significant increase in the expression of LgBiT and in the emission of luminescence in GBA1-HiBiT iPSCs and dNPCs (Figure 3E, Figure 4F–H,J and Figure 5A), it was not yet optimal in DNs for expression monitoring, so that we had to add LgBiT control protein to the cell lysate to detect a luminescence increase (Figure 5B). This implies that the *CLYBL* safe-harbor locus might be more silenced than expected upon differentiation into DNs, requiring additional optimization of the Dox dosing and potential addition of epigenetic regulators to enhance the LgBiT expression within the neurons themselves. One other caveat is that everything was done with cell lysates, meaning that the sensitivity of the system would be more optimal on a luminescent imaging platform, which was unavailable to us for these studies. Thus, further testing of LgBiT in DNs and other iPSC-derived cell types is still required.

In addition to the main master line, we characterized four iPSC clones that included at least one iLgBiT insertion, namely M1C6, M1D5, M1G4, and M2E3. The latter, as well as a fraction of M1G4, were confirmed with a single copy of LgBiT. M1C6 cells were suspected to be transformed with several copies of iLgBiT (Table 1). We did not investigate further how these cells could have been transformed, but we suspect that it may be due to the random insertion of the donor plasmid at off-target sites, causing genome alterations that may promote a growth advantage, confer an oncogenic state or disturb normal gene function. We decided not to pursue further studies with M1C6 to minimize the risks of insertional mutagenesis.

The rationale behind engineering a master inducible iLgBiT iPSC line was to generate isogenic pairs of DNs for the functional characterization of PD risk genes fused to HiBiT using luminescence microscopy. In this way, the power to detect significant phenotypic changes is increased by reducing variance through employing isogenic pairs, which relatively bypass the great complexity of interindividual human genetics [34].

The isolation of HiBiT-engineered iPSCs was performed without any antibiotic selection. Instead, we took advantage of a ddPCR method [34,35,36,37,38,39,40], and we adapted it to quantify in absolute mode the ratio of HiBiT-tagged alleles vs. an endogenous control gene. For each tested cell population, this quantification was performed through the simultaneous analysis of thousands of single-allele PCR reactions segregated into separate droplets. This was followed by limiting-dilution enrichment steps to isolate 100% HiBiT clones.

This HiBiT tagging system allows one to investigate proteins of interest, their localization, and expression levels without having to use specific antibodies whose syntheses require immunization and purification procedures [41]. This does not necessarily guarantee the specificity and sensitivity of the isolated antibodies. During the last decade, the scientific community has demonstrated a growing interest in validating the antibodies used in research [11,42], and procedures have been established to ensure the accuracy of antibody-dependent detected signals [10]. Nevertheless, many proteins remain difficult to study, either because of their conformation, their subcellular localization, and/or their low level of synthesis, which do not allow proper detection. In this study, we took advantage of the luminescence-based detection of the HiBiT/LgBiT system to overcome the limitation of detection observed with classical antibodies. Many proteins involved in PD, such as LRRK2 and PARKIN, encoded by genes that have been HiBiT-tagged in this study, were described with low expression levels in the dopaminergic lineage [43,44]. Those weak protein levels observed in rodents are in agreement with RNA sequencing data collected on human brain structure and referenced in the brainspan database (https://www.brainspan.org/, accessed on 30 June 2024) [45]. With that information in mind, we chose to implement the HiBiT/LgBiT system, which had already been described as highly sensitive [18,19], to detect and analyze the proteins involved in Parkinson’s disease.

In a regulatory context, HiBiT fusions in the N-terminal can be very useful for the detection of truncated proteins with potential residual and/or harmful activity, which might have escaped non-sense mediated decay (NMD) following indel/frameshift-induced knockouts (KOs) that rely on a premature termination codon (PTC). Such NHEJ-driven gene KOs are commonly achieved in preclinical studies. Most researchers frequently assume that their knockout is complete since the protein is not detectable using a specific antibody. However, some transcripts escaping NMD may be translated as truncated proteins and may not be recognized by any available antibody. Such peptides are undesirable in a KO setup since they can present some residual activity or cause other unintended damage [46,47]. To reveal the presence of any truncated form of proteins, the 33 bp HiBiT fragment can simply be flanked to the N-terminal part of the target protein using CRISPR-Cas9 editing, and luminescence can be quantified in the presence of the LgBiT subunit and the corresponding substrate.

We used an approach based on small molecules to differentiate iPSCs to reflect, as much as possible, the in vivo developmental trajectory of DNs. We demonstrated our ability to differentiate these LgBiT-HiBiT iPSCs into dNPCs and DNs as efficiently as with their isogenic M2E3 control. We believe that the various HiBiT fusions engineered in this work will help to shed light on the fundamental mechanisms underlying neuron degeneration in PD. This represents a power set of tools for the field towards exploring the localization and activity of PD-associated proteins across cell types and even more advanced 3D brain organoid models.

## 4. Materials and Methods

### 4.1. iPSC Culture

The use of human iPSCs in this study was approved by the Health Canada and Public Health Agency of Canada Research Ethics Board (REB 2022-001H). The AIW002-02 human iPSC line was reprogrammed from a 37-year-old Caucasian male and characterized as previously described [20]. In brief, peripheral blood mononuclear cells (PBMCs) were harvested, and Sendai viruses were used to overexpress *OCT4/POU5F1*, *SOX2*, *KLF4*, and *MYC* for cell reprogramming. Emerging iPSC colonies were selected for activated endogenous human pluripotency markers, differentiation potential into three germ layer cells after embryoid body formation in vitro, and normal karyotype. Short tandem repeat (STR) analysis was performed at The Centre for Applied Genomics (TCAG, Toronto, Canada) to authenticate tissue sample origins. Cells were maintained in a humidified chamber at 37 °C in 5% CO_2_ and normoxic conditions. Cells were cultured on Matrigel (Corning, Corning, New York, NY, USA, cat#354277) using mTeSRplus media (STEMCELL Technologies, Vancouver, BC, Canada, cat#100-0276) and passaged using ReLeSR (STEMCELL Technologies, cat#5872) according to the manufacturer’s directions. MycoZap Prophylactic antibiotic (Lonza, Basel, Switzerland, cat#VZA-2031) was used to prevent mycoplasma contamination during iPSC maintenance. Cells were routinely screened for mycoplasma contamination using a PCR detection kit (Abm, Cambridge, UK, cat#G238).

### 4.2. Karyotyping

Cells were cultured until they reached 50–60% confluence and were incubated for 30 min in Colcemid (0.15 µg/mL) to arrest the cell cycle at metaphase. They were rinsed with PBS and incubated in GCDR for 5 min. Cells were detached by scraping and then centrifuged for 5 min at 1000 rpm. The pellet was resuspended and incubated in 75 nM KCL hypotonic solution for 25 min at 37 °C. A few drops of Carnoy’s fixative were added and gently mixed by pipetting up and down. The mix was centrifuged at 1000 rpm for 10 min, the supernatant discarded, and the pellet resuspended, followed by the addition of 15 mL of Carnoy’s fixative. Slides were prepared by dipping in ethanol and placing in autoclaved cold water in a Coplin jar. Under the chemical fume hood, 2 drops of the suspended cells were added to the chilled slide, and a few drops of Carnoy’s fixative were added. The slide was dried at room temperature and then left overnight on a slide warmer (60–70 °C). The slides were dipped in a Coplin jar filled with 1% Trypsin in PBS. The slides were stained with Giemsa stain (1:4 dilution in Gurr buffer), rinsed with water, allowed to dry on a hotplate, and then analyzed under the microscope.

### 4.3. Generating a Doxycycline-Inducible LgBiT iPSC Line

Several modifications were applied to the original LgBiT DNA sequence (from Promega) before inserting it into the safe-harbor site CLYBL [21] of our human iPSC line AIW002-02. First, we designed a human codon-optimized version of LgBiT using the human codon optimization tool available from the Integrated DNA Technologies (IDT, Coralville, IA, USA) website. Second, we added a Kozak consensus sequence (GCCGCCACCATGG) at the beginning of the coding sequence, as well as a stop codon (TGA) at the end. Third, we flanked short homology arms (35–39 bp) for Gibson assembly. This new sequence was synthesized as a gBlock double-stranded DNA fragment (IDT) and subcloned into a Dox-inducible cassette. For this, Addgene plasmid #124229 (CLYBL-TO-hNGN2-BSD-mApple, gift from Michael Ward) was digested using the restriction enzymes BamH1 and Nru1 (New England Biolabs, Ipswich, MA, USA) to remove hNGN2 and linearize the vector. Our newly synthesized HA-Kozak-codon-optimized-LgBiT-stop-HA fragment (4 fmol) was ligated to the linearized vector (4 fmol) using a Gibson assembly kit (Thermo Fisher Scientific, Waltham, MA, USA). After the selection of transformed bacteria on Ampicillin (200 µg/mL; Fisher BP17605) agar plates, six colonies were screened, and all were revealed positive for the correct plasmid using agarose electrophoresis and Sanger sequencing. One of them was used to make a maxiprep (ThermoFisher) of the resulting plasmid, which was named ‘’pCLYBL-iLgBiT’’. This plasmid was used for CRISPR-Cas9 editing experiments as a template for homology-directed repair (HDR)-based insertion of iLgBiT into the safe-harbor locus within exon 2 of the human gene CLYBL.

The sgRNAs were designed using tools available at benchling.com. For the insertion of iLgBiT into CLYBL, 1.2 µL of a synthetic sgRNA (Synthego, Redwood, CA, USA; 10 µM stock; Table 2) was mixed with 2 µL of Cas9 protein (ThermoFisher; 1 mg/mL stock). After a 10-min incubation time at room temperature (RT) to form a Cas9: sgRNA ribonucleoprotein (RNP) complex, 2 µg of plasmid pCLYBL-iLgBiT was added to the RNP complex, as well as 0.5 µg of Addgene plasmid # 41856 (pCE-mp53DD, gift from Shinya Yamanaka; dominant-negative p53) to improve survival of iPSCs affected by Cas9-induced double-stranded breaks. This RNP/plasmids mixture was transfected into 1.5 × 10^6^ AIW002-02 iPSCs, freshly dissociated with Accutase (STEMCELL Technologies, cat#07920), using Lipofectamine Stem (ThermoFisher, STEM00001) into one well of a 6-well plate, previously coated with Matrigel (Corning, cat#354277), containing 2 mL of mTeSR1 (STEMCELL Technologies, cat#85850) and 10 µM Rock inhibitor (STEMCELL Technologies, cat#72304). After 24 h, the medium was replaced with 2 mL of fresh mTeSR supplemented with 10 µM Rock inhibitor.

The following day, the medium was replaced with 2 mL of mTeSRplus, without Rock inhibitor, and 25 µg/mL of Neomycin (Gibco, ThermoFisher, Waltham, MA, USA, cat#10131-035). The same media was renewed every two days. We kept these iPSCs under neomycin selection for three weeks with regular cell passaging every time they reached 50% confluency. After Neomycin selection, iPSCs were dissociated with Accutase, and 100,000 cells were seeded in each well of a 6-well plate containing 2 mL mTeSR1 with 10 μM Rock inhibitor. Twenty-four hours later, the medium was changed to mTeSR1 containing 2 μM TAT-Cre recombinant protein (EMD Millipore, Burlington, VT, USA, cat#SCR508) to trigger the excision of the floxed Neomycin selection cassette. Cells were incubated with TAT-Cre for 24 h. Then, cells were washed with PBS without calcium or magnesium (Corning, cat#21-040-CV) and fed with 2 mL mTeSRplus. Cells were maintained until confluency with media changes every two days and further expanded for downstream validation.

The next step in our workflow was to generate monoclonal populations of the engineered cells before further screening and validation. Fluorescence-activated cell sorting (FACS; BD FACSAria Fusion, BD Biosciences, San Jose, CA, USA) was used to deposit single cells into each well of a 96-well plate pre-coated with Matrigel. Briefly, cells were treated with 10 µM of Rock Inhibitor for two hours prior to sorting. Cells were dissociated with Accutase for 10 min at 37 °C, then rinsed with 4 mL of mTeSRplus media and were pelleted using 250 g for 3 min. Cells were later resuspended in mTeSRplus media supplemented with 10 µM of Rock inhibitor, passed through a 35 µM strainer, immediately sorted as single cells into 96-well plates pre-coated with Matrigel, and loaded with 100 µL per well of mTeSRplus media supplemented with 10 µM of Rock inhibitor. After 48 h, cells were fed with fresh mTeSRplus media and maintained until colonies emerged and were ready for passaging in 10–14 days. This method yields between 10 and 20 clones per 96-well plate.

Single cell-derived colonies were screened to detect the presence of the iLgBiT cassette using ddPCR. For the TaqMan^®^ assay underlying the ddPCR absolute quantification mode, custom primers to amplify the iLgBiT allele (CLYBL-LgBiT-taqF and CLYBL-LgBiT-taqR), the control SYT1 allele (SYT1-Ftaq and SYT1-Rtaq; Table 2), as well as DNA locked nucleic acid (LNA^®^, Thermo Fisher Scientific) probes, fused to different fluorophores (FAM and HEX), were designed following stringent criteria previously described [35]. One probe was specific to the iLgBiT allele (CLYBL-LgBiT-FAM), and the other probe was to our control endogenous autosomal gene SYT1 (SYT1-wtHEX; Table 2). Clones carrying an allele ratio iLgBiT:SYT1 of 0.5 (heterozygous) were preferentially submitted to a subsequent PCR step to confirm the removal of the Neomycin selection cassette from the CLYBL target locus by Cre recombinase using different primer sets, i.e., CLY5-F2 and Neo-R2, to show the presence of Neomycin selection cassette, as well as CLY5-F1 and Cre-R1 to reveal the absence of Neomycin selection cassette, at the same locus (Table 2). Moreover, an additional primer set was designed to reveal any wt CLYBL allele in which no insertion had been achieved at the same target site using CLYwt-F2 and CLYwt-R2 (Table 2). Isolated clones were assessed for iLgBiT sequence integrity by PCR cloning and Sanger sequencing using the primers M13rev and LNCX (Table 2).

### 4.4. Generation of HiBiT iPSC Lines

To knock in the HiBiT sequence to our selected target genes, 1 µL of Cas9 protein (IDT; stock 61 µM) was mixed with 3 µL of synthetic sgRNA (Synthego; stock 100 µM; Table 3) at RT for 10–20 min to form a Cas9: sgRNA ribonucleoprotein (RNP) complex. After formation of the RNP complex, 1 µL of ssODN (IDT; stock 100 µM; Table 3) and 20 µL of nucleofection buffer P3 (Lonza) were added to the RNP mix. M2E3 iPSCs at a maximum of 50% confluency were washed with PBS, and detached with Accutase at 37 °C for 10 min. After centrifuging 500,000 detached cells in a separate tube at 250 g for 3 min, cells were washed with 5 mL PBS, centrifuged again, and the pellet was resuspended gently with 25 µL of the RNP-ssODN-buffer mix. Cells were nucleofected using the program CA137 in a Nucleofector 4D device (Lonza) and plated onto one Matrigel-coated 10 cm tissue culture dish including 10 mL mTeSR1, ROCK inhibitor 10 µM, and HDR enhancer (IDT) 30 µM. The day after, media was replaced by mTeSRplus. After allowing sufficient time for recovery and initial growth, individual colonies were manually isolated under a microscope. Each picked colony was carefully transferred into one well of a Matrigel-coated 96-well plate, ensuring that each well contained a single colony. When 80–90% confluent, cells were washed with PBS and detached with 30 μL/well of Accutase for 10 min at 37 °C. Half of each well (15 µL) was transferred into a new Matrigel-coated 96-well plate pre-filled with 150 μL/well of mTeSRplus and Rock inhibitor 10 µM and placed in the incubator at 37 °C as a backup plate. We added 35 µL/well of QuickExtract™ DNA Extraction Solution (Lucigen, Middleton, WI, USA, cat#QE09050) to the remaining cells/Accutase mix in the original plate and heated at 65 °C for 10 min, then 95 °C for 5 min. The DNA extract was diluted 1/20 with H_2_O to evaluate the proportion of edited alleles using ddPCR. For each 96-well plate, we mixed 1,170 µL of ddPCR Supermix for Probes (no dUTP; BioRad, Hercules, CA, USA, cat#1863024), 975 µL H_2_O, 20.8 µL of primers (IDT, stock 100 µM; Table 3) and 6.5 µL of probe (IDT, stock 100 µM; Table 3), and distributed 19 µL per well into a new ddPCR 96-well plate (BioRad, 12001925). We added 2 µL per well of the diluted DNA extract into each corresponding well. The droplets were assembled using the QX200™ Droplet Generator (BioRad, cat#1864002) according to the manufacturer’s protocol and transferred into a ddPCR 96-well plate. The PCR reaction was performed using the C1000 Touch™ thermal cycler with 96–deep well reaction module (BioRad, cat#1851197). The resulting droplets were analyzed using the QX200™ Droplet Reader (BioRad, cat#1864003), and the absolute quantification of edited (HiBiT knockin) and unedited alleles was obtained. If no well was identified with 100% edited alleles, we searched for the well(s) with the highest proportion of edited alleles. When the corresponding well in the backup plate reached 60–70% confluency, cells were washed with PBS and treated with 30 μL per well of Accutase at 37 °C until all cells lifted off. We added 250 µL of mTeSR1 and Rock inhibitor 10 µM per well and mixed gently by pipetting up and down 2–3 times, counted the cells, transferred 100–200 cells into a new 15 mL conical tube previously filled with 10 mL of mTeSR1 and Rock inhibitor 10 µM, and distributed 100 µL/well of this mix in a new Matrigel-coated 96-well plate. This enrichment step was repeated until a well with 100% edited cells was found.

Isolated iPSC clones were assessed for sequence integrity, potential off-target editing events, and homozygosity/heterozygosity of HiBiT insertion by PCR cloning and Sanger sequencing. We designed PCR primers (Table 3) that gave rise to 366–662 bp single/clean bands on an agarose gel with the HiBiT sequence located in the middle. Each fragment was used for PCR cloning or directly for Sanger sequencing using the same PCR primers.

### 4.5. Doxycycline-Induction of iLgBiT Expression

To induce iLgBiT expression in our engineered cells, doxycycline hyclate (Thermo Fisher Scientific, cat#AC446060050) was added to the culture media at a concentration of 2 µg/mL for 48 h. Note that it was important to ensure that no MycoZap antibiotic was added to the media for a minimum period of 3 days prior to the assay since this system uses Tet Response Elements (TREs) and MycoZap can mimic Dox effects. The expression of iLgBiT was monitored at both transcript and protein levels.

### 4.6. RT-qPCR

For iLgBiT transcript levels, the bench and equipment were first cleaned with RNaseZap Wipes (Ambion, Thermo Fisher Scientific, AM9786) or RNASE AWAY (MolecularBioProducts, San Diego, CA, USA, cat#7003) prior to starting any work. Culture media was discarded from each well of a near-confluent 6-well plate, and cells were washed 2 times with 2 mL PBS. All remaining PBS was aspirated, and cells were treated with 1 mL Accutase at 37 °C until detachment, usually for 10 min. The well content was transferred to an RNase-free 2 mL polypropylene centrifuge tube. Dissociated cells were pelleted at 250 g for 3 min, and the supernatant was removed. Cells were rinsed with 2 mL PBS, pelleted again, and supernatant completely removed. Total RNA was purified from cells using gDNA Eliminator columns from the RNeasy Plus Mini Kit (Qiagen, Hilden, Germany, cat#74134) according to the manufacturer’s instructions. The concentration of the extracted RNA samples was measured on a NanoDrop spectrophotometer, and first-strand cDNA synthesis was performed using the SuperScript III First-Strand Synthesis System for RT-PCR (Invitrogen, Thermo Fisher Scientific, cat#18080-051). Each cDNA sample was then submitted to RT-qPCR (Applied Biosystems, Thermo Fisher Scientific) in quadruplicates to assess the transcript levels using TaqmanTM assays or PowerUp™ SYBR™ Green Master Mix (Thermo Fisher Scientific, cat#A25742) in combination with custom primers (Table 4), according to the manufacturer’s instructions, in an optical plate (Applied BioSystems, cat#4346906) sealed with an optical cover (Applied Biosystems, cat#4311971).

### 4.7. Induction and Differentiation of iPSC-Derived Dopaminergic Progenitors and Neurons

The iPSC culture and generation of progenitors were adapted from published methods [20,38]. Briefly, iPSCs were cultured on mTeSR1 for at least two passages and seeded as a single-cell suspension in a microfabricated embryoid body disk device (EB disk, eNUVIO, Montreal, Canada) to generate EBs. Following a week of culture in dopaminergic EB media, the EBs were replated and induced into neural rosettes using dopaminergic induction media. Dopaminergic progenitors were then obtained from neural rosettes that were cultured in dopaminergic progenitor culture media. Finally, progenitors were cultured for one week in differentiation media to obtain 1 week-old dopaminergic neurons.

### 4.8. Ambroxol Treatment

Ambroxol hydrochloride (Sigma-Aldrich, St. Louise, MO, USA, cat#A9797) was dissolved in DMSO by vigorous shaking and sterilized by filtration with 0.2 µm filter. Fresh dopaminergic progenitor media supplemented with 70 µM Ambroxol was added to each well (2 mL/well, 6-well plate). Media with Ambroxol was changed every 48 h over 6 days. Control cells were treated with 0.1% DMSO.

### 4.9. Western Blot

Cells were rinsed with PBS and incubated in Accutase for 5 min at 37 °C. To quench the enzymatic reaction, DMEM/F12 was added 1:1. The cell suspension was centrifuged at 4 °C for 5 min and rinsed with cold PBS. The cell pellet was lysed by resuspension in RIPA buffer containing a protease and phosphatase inhibitor cocktail (Roche, Basel, Switzerland). After 20 min on ice, the lysates were centrifuged for 15 min at 12,000× *g*. The protein extracts were transferred to a fresh tube and stored at −20 °C. For sample preparation, proteins were quantified using the BioRad DC Protein assay (BioRad). The protein lysates were mixed with 4x Laemmli buffer and boiled for 5 min at 95 °C. Equal amounts of protein (20 µg) were loaded for each sample and electrophoresed on a 12% polyacrylamide gel. Following SDS-PAGE electrophoresis, proteins were transferred onto nitrocellulose membranes via Trans-Blot Turbo Transfer system at 2.5 mV, 10 min (BioRad). The membrane was blocked in 5% BSA or milk diluted in Tris-buffered saline containing 0.1% Tween 20 (TBST) or overnight at room temperature for 1 hr. Membranes were incubated with primary antibodies diluted in blocking buffer overnight at 4 °C (Table 5). Following washing in 0.1% TBST, the membranes were incubated with peroxidase secondary antibodies diluted in blocking buffer for 1 h at room temperature. Membranes were washed again and revealed by Clarity Western ECL Substrate (BioRad). Image acquisition and densitometry were performed with ChemiDoc MP System (BioRad) and ImageJ.

### 4.10. Luminescence Detection

The luminescence production was monitored using the Nano-Glo Lytic Detection kit (Promega, Madison, WI, USA, cat#N3030). For one well of a 6-well plate, cells were washed with 1 mL PBS, then 1.01 mL lysing solution was added (composed of 500 µL Nano-Glo HiBiT lytic buffer, 10 µL Nano-Glo HiBiT lytic substrate, and 500 µL of PBS). Cells were incubated at room temperature for 10 min, protected from light while shaking gently, then deposited in 100 µL aliquots into white opaque 96-well plates (Thermo Fisher Scientific, 136101). HiBiT control protein (Promega, N3010) or LgBiT control protein (Promega, N401A; 1/100) were added to the cell lysates when required, followed by incubating the cells for 10 min in the dark at room temperature while shaking. Luminescence was measured using a luminometer (Synergy Mx, BioTek, Winooski, VT, USA). Note that to prevent light cross-contamination and obtain better results, at least one well was left empty between each sample.

### 4.11. GCase Enzyme Activity Assay

The GCase activity was measured using an artificial substrate of GCase, 4-methylumbelliferyl-β-D-glucopyranoside (4-MUG, Sigma-Aldrich, cat#M3633), that releases fluorogenic byproduct 4-methylumbelliferone (4-MU) when it is catalyzed by the enzyme measurable by a microplate reader. The GCase assay was performed according to methods published [48]. Briefly, the protein samples (5–10 µg) were diluted with RIPA buffer to final volume of 20 µL and loaded to a 96-well black plate on ice. We then applied 40 µL of reaction master mix consisting of GCase assay, 10% BSA, 2.5 mM 4-MUG, and 25 mM CBE (Sigma-Aldrich, cat#C5424) or distilled water. Using SpectraMax iD5 (Molecular Devices, San Jose, CA, USA), the fluorescence intensity was measured every 2 min with 30 s shaking between cycles for 2.5 h at 37 °C. For analysis of enzyme kinetics, Vmax values for each sample were normalized to the protein concentration.

### 4.12. Immunocytochemistry Analysis

Cells were fixed in 4% paraformaldehyde (PFA) in phosphate-buffered saline (PBS) at room temperature for 15 min, permeabilized with 0.2% Triton X-100/PBS for 15 min at RT, then blocked in 1% BSA, 5% donkey serum, and 0.05% Triton X-100/PBS for 1 h or overnight at 4 °C. Cells were incubated with primary antibodies (Table 5) in blocking buffer overnight at 4 °C on a shaker. Secondary antibodies were applied for 2 h at RT, followed by Hoechst (Thermo Fisher Scientific, cat#H3570) counterstain for 5 min. Images were acquired using the automated Image Xpress imaging system (Thermo Fisher Scientific). Images were processed and analyzed using ImageJ software (NIH, Bethesda, MD, USA, version 1.53j).

### 4.13. Off-Target Analysis of LgBiT and HiBiT Insertions

To screen the genomic sites that are most likely to be targeted by off-target activity for any sgRNA used in this study, Benchling.com was used to predict and locate the top three off-target candidate loci based on the highest score (mainly lowest number of mismatches compared with the sgRNAs used) for both LgBiT and HiBiT insertions. Primers (Table 6) were designed to flank each of these regions, and PCR-amplified regions were further analyzed by Sanger sequencing to confirm the integrity around the region containing the predicted cut site in the event of off-target activity, which is three base pairs upstream of the PAM site NGG.

### 4.14. Statistical Analysis

Statistical tests and figures were generated using GraphPad Prism software (San Diego, CA, USA, version 10). An unpaired *t*-test was used for comparison between groups, while one- and two-way ANOVA with Bonferroni tests were used for comparison between multiple groups. Values are presented as mean ± SD. Asterisks in the figures denote statistical significance with the highest *p*-value threshold used as the significance indicator.

## Figures and Tables

**Figure 1 ijms-25-09493-f001:**
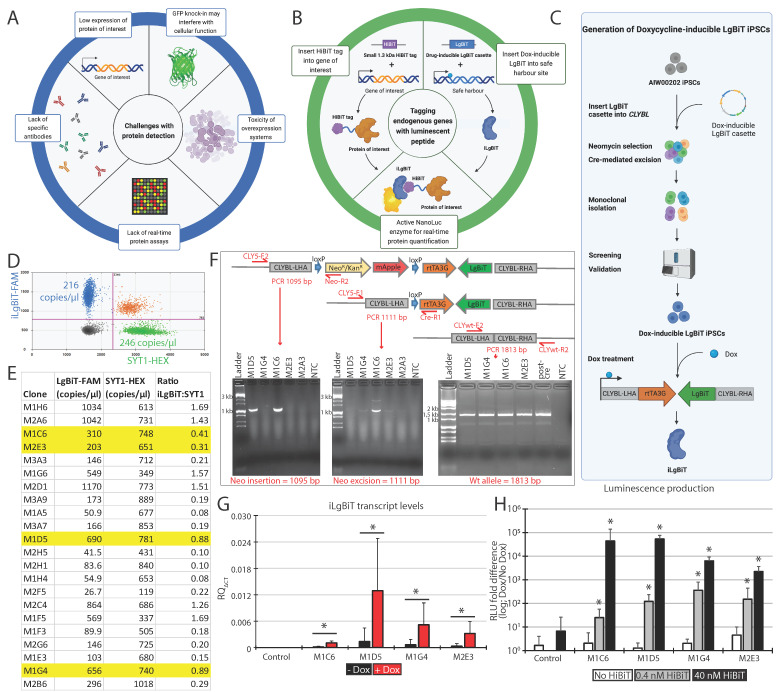
Generation of the master Dox-inducible LgBiT iPSC line. (**A**) Examples of challenges faced by current protein detection tools. (**B**) Overview of the LgBiT-HiBiT luminescent detection system. (**C**) Outline of the method used to generate the Dox-inducible LgBiT iPSCs. (**D**) Representative 2D ddPCR profile of an edited iPSC population composed of a ratio of 0.88 of CLYBL-iLgBiT alleles (blue) vs. SYT1 alleles (endogenous control; green). (**E**) Calculation table of the ratio iLgBiT:SYT1 alleles in different edited iPSC clones. (**F**) Top: graphical representation of the Dox-inducible expression cassette inserted into intron 2 (safe-harbor locus) of *CLYBL*, before and after treatment with Cre recombinase to remove the Neomycin selection cassette, as well as the *CLYBL* wt allele (without any insertion at the safe-harbor site); bottom: electrophoresis analysis of the corresponding PCR fragments. (**G**) Transcript levels of iLgBiT in different edited iPSC clones before (black) and after (red) Dox treatments. Values are presented as mean ± SD of (n) independent experiments, where n = 3 for M1G4 and M2E3 and n = 2 for M1C6 and M1D5, each including three technical replicates. Two-tailed paired *t*-test was used to calculate the p-values between each Dox-treated and non-treated sample. (**H**) Luminescence assay evaluating the expression of LgBiT upon Dox treatment in our different engineered iPSC lines, expected to reconstitute the nanoluciferase activity in the presence of different concentrations of HiBiT control protein added to the lysates. Values are presented as mean ± SD of three independent experiments, each including two technical replicates. Two-tailed paired *t*-test was used to calculate the *p*-values between each Dox-treated and non-treated sample. * *p* < 0.03; LHA: left homology arm; RHA: right homology arm; Neo^R^: Neomycin resistance; Kan^R^: Kanamycin resistance; wt: wild-type.

**Figure 2 ijms-25-09493-f002:**
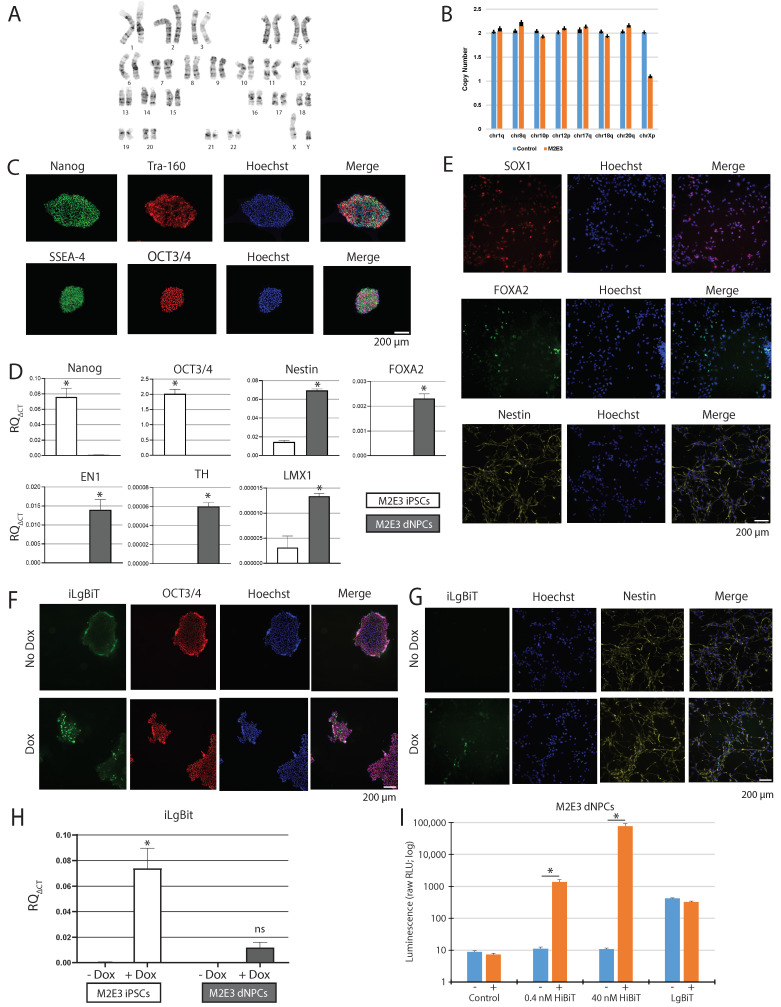
Characterization of M2E3 iPSCs and iPSC-derived dNPCs. (**A**) G-band analysis and (**B**) qPCR-based stability test revealed normal karyotype. (**C**) ICC detection of Nanog, Tra-160, SSEA-4, and OCT3/4 in M2E3 iPSCs. Scale bar 200 μm. (**D**) Transcript levels, evaluated by RT-qPCR, of Nanog, OCT3/4, Nestin, FOXA2, EN1, TH, and LMX1 in M2E3 iPSCs and dNPCs. Values are presented as mean ± SD of three experiments, each including three technical replicates. Two-tailed paired *t*-test was used to compare iPSC and dNPC samples; * *p* < 0.002. (**E**) ICC detection of SOX1, FOXA2, and Nestin in M2E3 dNPCs. (**F**) ICC co-staining of iLgBiT with OCT3/4 in M2E3 iPSCs, and (**G**) iLgBiT with Nestin in M2E3 dNPCs, treated or not with Dox. Scale bar 200 μm. (**H**) Transcript levels, evaluated by RT-qPCR, of iLgBiT in M2E3 iPSCs and dNPCs, treated or not with Dox. Values are presented as mean ± SD of three experiments, each including three technical replicates. Two-way ANOVA followed by Bonferroni test was used to compare Dox and no Dox; * *p* < 0.002. (**I**) Bar graph showing luminescence assay readout in M2E3 dNPCs, with 0 nM, 0.4 nM, or 40 nM HiBiT control protein, or LgBiT control protein 1/100, added to the lysates, after treatment with Dox or no. Relative light unit (RLU) values are presented as mean ± SD of 2 independent experiments, each including 2 technical replicates; * *p* < 0.05.

**Figure 3 ijms-25-09493-f003:**
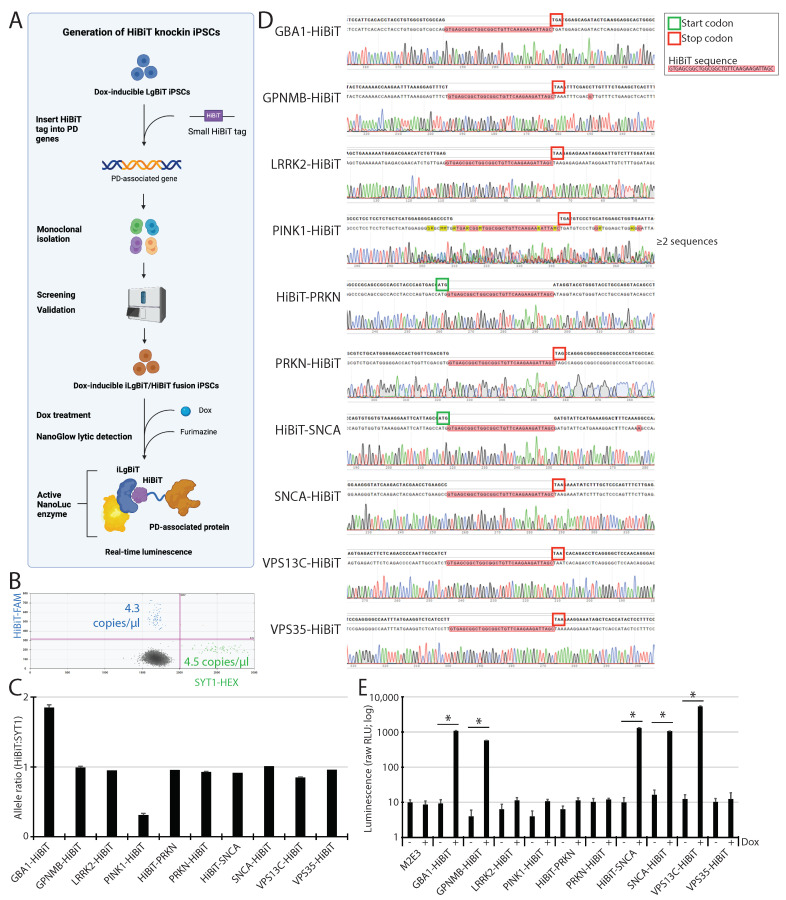
Generation of the HiBiT knock-in iPSC lines. (**A**) Outline of the method used to generate HiBiT-tagged genes in Dox-inducible LgBiT iPSCs, in order to produce luminescence. (**B**) Representative 2D ddPCR profile of a HiBiT knock-in iPSC population composed of a ratio ~1.0 of HiBiT alleles (blue) vs. SYT1 alleles (endogenous control; green). (**C**) Bar graph presenting the allele ratio HiBiT:SYT1, as measured by ddPCR, for each HiBiT knock-in iPSC clone isolated. Values are presented as mean ± SD of one experiment, including five technical replicates for GBA1-HiBiT, three for GPNMB-HiBiT, PINK1-HiBiT, PRKN-HiBiT, VPS13C-HiBiT, and one for the others. For each HiBiT-tagged line, between 24 and 32 separate clones were screened before finding at least one fully edited. (**D**) Sanger sequencing chromatograms depicting the insertion site of HiBiT (pink sequence), right before the stop codon (red frame), or right after the start codon (green frame), for each HiBiT knock-in iPSC clone isolated. Note that PINK1-HiBiT is the only one with an unclear sequence profile, probably composed of one HiBiT allele and one wt and/or indel allele. (**E**) Bar graph showing luminescence assay readout in master M2E3 iPSC line, as well as in the different HiBiT knock-in iPSC lines, treated (+) or not (−) with Dox. Raw relative light units (RLU) values are presented as mean ± SD of one experiment, including 3 technical replicates. Two-tailed paired *t*-test was used to calculate the *p*-values between each Dox-treated and non-treated samples; * *p* < 0.01.

**Figure 4 ijms-25-09493-f004:**
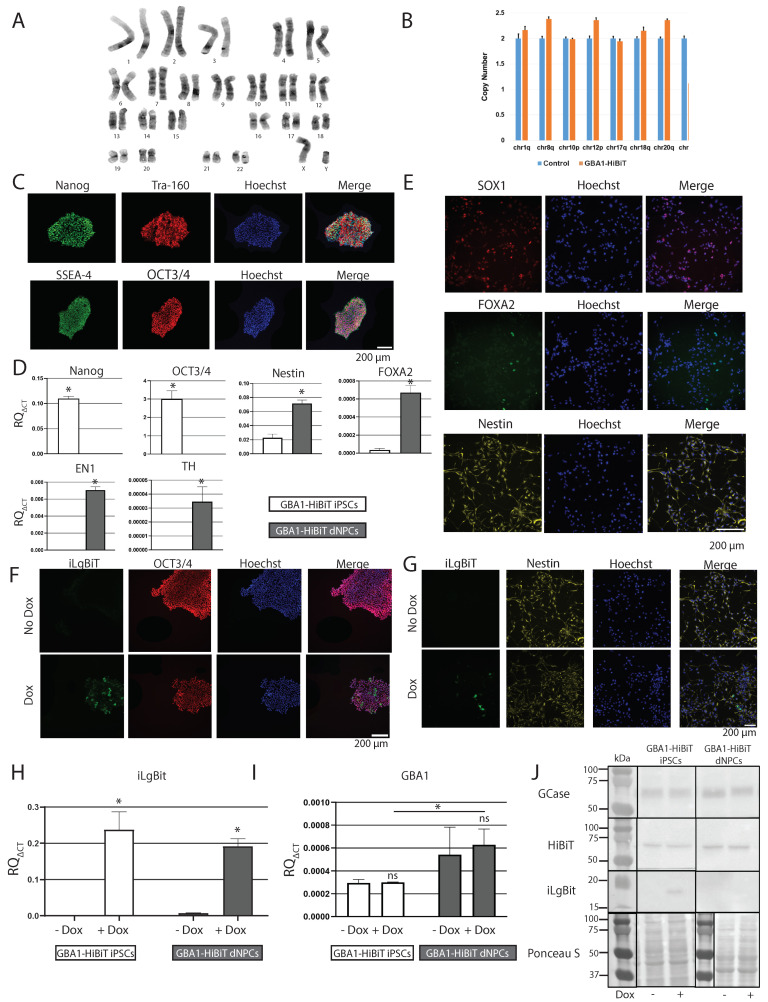
Characterization of GBA1-HiBiT iPSCs and iPSC-derived dNPCs. (**A**) G-band analysis and (**B**) qPCR-based stability test revealed normal karyotype. (**C**) ICC detection of Nanog, Tra-160, SSEA-4, and OCT3/4 in GBA1-HiBiT iPSCs. Scale bar 200 µm. (**D**) Transcript levels, evaluated by RT-qPCR of Nanog, OCT3/4, Nestin, FOXA2, EN1, and TH in GBA1-HiBiT iPSCs and dNPCs. Values are presented as mean ± SD of three experiments, each including three technical replicates. Two-tailed paired *t*-test was used to compare iPSC and dNPC samples; * *p* < 0.0001. (**E**) ICC detection of SOX1, FOXA2, and Nestin in GBA1-HiBiT dNPCs. (**F**) ICC co-staining of iLgBiT with OCT3/4 in GBA1-HiBiT iPSCs, and (**G**) iLgBiT with Nestin in GBA1-HiBiT dNPCs, treated or not with Dox; scale bar 200 µm. (**H**) Transcript levels evaluated by RT-qPCR of iLgBiT in GBA1-HiBiT iPSCs and dNPCs, treated or not with Dox. Values are presented as mean ± SD of three experiments, each including three technical replicates. Two-way ANOVA followed by Bonferroni test was used to compare Dox and no Dox; * *p* < 0.0001. (**I**) Transcript levels, evaluated by RT-qPCR of GBA1-HiBiT in GBA1-HiBiT iPSCs and dNPCs, treated or not with Dox. Two-way ANOVA followed by Bonferroni test. Values are presented as mean ± SD of three experiments, each including three technical replicates. Two-way ANOVA followed by Bonferroni test was used to compare Dox and no Dox; * *p* < 0.05. (**J**) ICC detection of GCase, HiBiT, iLgBiT, and GAPDH in GBA1-HiBiT iPSCs and dNPCs, treated or not with Dox.

**Figure 5 ijms-25-09493-f005:**
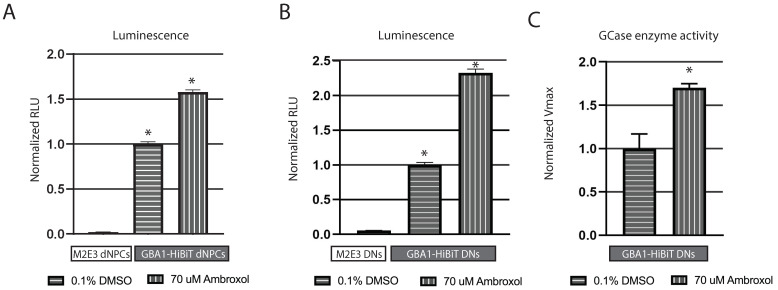
Functional validation of LgBiT-HiBiT system with a chaperone of GCase, Ambroxol, in M2E3 and GBA1-HiBiT, dNPCs, and DNs. (**A**) Luminescence measurement in M2E3 and GBA1-HiBiT dNPCs and (**B**) in 1-week DNs following a 6-day treatment with 70 μM Ambroxol. iLgBiT expression was induced by a 2-day Dox treatment in dNPCs and DNs and the addition of LgBiT recombinant protein in DNs. Relative light unit (RLU) values were normalized to DMSO conditions and presented as mean ± SD of four independent experiments, each including two technical replicates. Statistical comparisons were made using one-way ANOVA followed by Bonferroni test; * *p* < 0.03. (**C**) 4-MUG kinetic assay for GCase enzyme activity in GBA1-HiBiT 1-week old DNs, treated or not with 70 μM Ambroxol for 6 days. Values are presented as mean ± SD of two independent experiments, each including two technical replicates. Two-tailed paired *t*-test was used for comparison; * *p* < 0.03.

**Table 1 ijms-25-09493-t001:** Evaluation of the allele ratio iLgBiT:SYT1 by ddPCR from different subclones derived from clones M1C6, M2E3, and M1G4.

	Clone	LgBiT-FAM (copies/µL)	SYT1-HEX (copies/µL)	Ratio iLgBiT:SYT1	Clone	LgBiT-FAM (copies/µL)	SYT1-HEX (copies/µL)	Ratio iLgBiT:SYT1	Clone	LgBiT-FAM (copies/µL)	SYT1-HEX (copies/µL)	Ratio iLgBiT:SYT1
	M1C6.1	525	193	2.7	M2E3.v3-1	99	192	0.5	M1G4.v3-1	78	175	0.4
	M1C6.2	326	88	3.7	M2E3.v3-2	29	60	0.5	M1G4.v3-2	936	377	2.5
	M1C6.3	295	89	3.3	M2E3.v3-3	135	271	0.5	M1G4.v3-3	104	213	0.5
	M1C6.4	615	231	2.7	M2E3.v3-4	96	146	0.7	M1G4.v3-4	67	158	0.4
	M1C6.5	612	191	3.2	M2E3.v3-5	75	155	0.5	M1G4.v3-5	197	466	0.4
	M1C6.6	425	120	3.5	M2E3.v3-6	38	72	0.5	M1G4.v3-6	181	413	0.4
	M1C6.7	671	207	3.2	M2E3.v3-7	64	122	0.5	M1G4.v3-7	130	307	0.4
	M1C6.8	409	119	3.4	M2E3.v3-8	56	146	0.4	M1G4.v3-8	25	52	0.5
	M1C6.9	499	146	3.4	M2E3.v3-9	78	152	0.5	M1G4.v3-9	130	285	0.5
	M1C6.11	523	169	3.1	M2E3.v3-10	96	204	0.5	M1G4.v3-10	111	231	0.5
	M1C6.12	464	129	3.6	M2E3.v3-11	95	209	0.5	M1G4.v3-11	112	253	0.4
	M1C6.13	525	187	2.8	M2E3.v3-12	88	177	0.5	M1G4.v3-12	243	506	0.5
	M1C6.14	305	105	2.9	M2E3.v3-13	73	131	0.6	M1G4.v3-13	201	458	0.4
	M1C6.15	216	246	0.9	M2E3.v3-14	22	56	0.4	M1G4.v3-14	1663	633	2.6
					M2E3.v3-15	109	241	0.5	M1G4.v3-15	89	32	2.8
					M2E3.v3-16	127	287	0.4	M1G4.v3-16	199	482	0.4
Ave				3.0				0.5				0.9
SD				0.7				0.07				0.9

Note: Ave = average; SD = standard deviation.

**Table 2 ijms-25-09493-t002:** Nucleotide sequence of primers and probes used for characterization of LgBiT knock-in into *CLYBL*.

Oligonucleotide Name	Oligonucleotide Sequence
CLYBL-sgRNA	ATGTTGGAAGGATGAGGAAA
CLYBL-LgBiT-taqF	CATCTGATCGGCAGAGAGC
CLYBL-LgBiT-taqR	ACAAGGTGGCGTCAGTTC
SYT1-Ftaq	AGCCATAGTCGCAGTCCT
SYT1-Rtaq	ACCTGATCTTTCATCGTCTTCC
CLYBL-LgBiT-FAM	CCA+GA+C+C+GGA
SYT1-wtHEX	AA+GAA+GAA+G+G+GA
CLY5-F2	TCCTACTGGAGACACAGGTCC
Neo-R2	TCCACGTCACCGCATGTTAG
CLY5-F1	GCTTGCACGTCTGGAACTCT
Cre-R1	GGCAAACAACAGATGGCTGG
CLYwt-F2	AAAACAGCATGACTGGTGGC
CLYwt-R2	ACCTGACGTTTTTCTACTGGGA
M13rev	CAGGAAACAGCTATGAC
LNCX	AGCTCGTTTAGTGAACCGTCAGATC

Note: + = locked nucleic acid (LNA).

**Table 3 ijms-25-09493-t003:** Nucleotide sequence of gRNA, ssODN, primer, and probes used for characterization of HiBiT knock-in into different target genes.

Gene Fusion	Transcript ID	Oligonucleotide Name	Oligonucleotide Sequence
HiBiT-SNCA	ENST00000394991.8	gRNA	GTAAAGGAATTCATTAGCCA
		ssODN	TTCTCATTCAAAGTGTATTTTATGTTTTCCAGTGTGGTGTAAAGGAATTCATTAGCCATGGTGAGCGGCTGGCGGCTGTTCAAGAAGATTAGCGATGTATTCATGAAAGGACTTTCAAAGGCCAAGGAGGGAGTTGTGGCTGCTGCTGAGAAA
		Ftaq	TGTATTTTATGTTTTCCAGTGTGGT
		Rtaq	GCCACACCCTGTTTGGT
		wt-HEX	CA+T+G+G+AT+G+TATTCA
		HiBiT-FAM	CA+T+GG+TGA+G+CG
		Fseq	TCCGTGGTTAGGTGGCTAGA
		Rseq	CCATCACTCATGAACAAGCACC
SNCA-HiBiT	ENST00000394991.8	gRNA	TGGGAGCAAAGATATTTCTT
		ssODN	GTGCTGTCTTTTTGATTTTTCTAATATTAGGAAGGGTATCAAGACTACGAACCTGAAGCCGTGAGCGGCTGGCGGCTGTTCAAGAAGATTAGCTAAGAAATATCTTTGCTCCCAGTTTCTTGAGATCTGCTGACAGATGTTCCATCCTGTACAA
		Ftaq	TTAGGAAGGGTATCAAGACTACG
		Rtaq	GGCACATTGGAACTGAGC
		wt-HEX	CT+G+AAG+C+C+T+AAG
		HiBiT-FAM	CTG+G+CG+G+CTG
		Fseq	GTGCATCCGGATCAGAACCTA
		Rseq	CAGTGAAAGGGAAGCACCGA
LRRK2-HiBiT	ENST00000298910.12	gRNA	CTGTTGAGTAAGAGAGAAAT
		ssODN	GAAAAACACATTGAAGTGAGAAAAGAATTAGCTGAAAAAATGAGACGAACATCTGTTGAGGTGAGCGGCTGGCGGCTGTTCAAGAAGATTAGCTAAGAGAGAAATACGAATTGTCTTTGGATAGGAAAATTATTCTCTCCTCTTGTAAATATTTATTTTAAA
		Ftaq	ATCAATCTTCCACATGAAGTGC
		Rtaq	ATGTGAGTACCCTTTCCATGT
		wt-HEX	T+C+T+GTT+G+A+GTAAGA
		HiBiT-FAM	CGG+CTGG+CGG
		Fseq	ACTAAAAATACATGAGCCAAACTGA
		Rseq	ACCTCCATTACAGACAAGAAAACA
PINK1-HiBiT	ENST00000321556.5	gRNA	GCTCCATGCAGGGACATCAC
		ssODN	TCTGCCAGGCAGCCCTCCTCCTCTGCTCATGGAGGGCAGCCCTGGTGAGCGGCTGGCGGCTGTTCAAGAAGATTAGCTGATGTCCCTGCATGGAGCTGGTGAATTACTAAAAGAACATGGCATCCTCTGTG
		Ftaq	CTGGAGTGTGAAACGCTCT
		Rtaq	CACAGACCATCACGACACA
		wt-HEX	AG+CC+CTGT+G+ATG
		HiBiT-FAM	A+G+ATTA+G+C+T+GATGT
		Fseq	AGACCCTCACTAACAAAGCAGG
		Rseq	TTCTTCCATTTGCCAAGCCC
GPNMB-HiBiT	ENST00000258733.9	gRNA	AAAAGTGAGCTTCAGAAACA
		ssODN	TTCCCGGGAAACCAGGAAAAGGATCCGCTACTCAAAAACCAAGAATTTAAAGGAGTTTCTGTGAGCGGCTGGCGGCTGTTCAAGAAGATTAGCTAAATTTCGACGTTGTTTCTGAAGCTCACTTTTCAGTGCCATTGATGTGAGATGTGCTGGAGTGGCTATTAACCTT
		Ftaq	CCGGGAAACCAGGAAAAGG
		Rtaq	TCAACTTCCCCAAACCACAA
		wt-HEX	A+G+G+A+GTT+T+CTTAAA
		HiBiT-FAM	CG+GC+TGG+CGG
		Fseq	CCAGTGTCTTGCAAACTGTCAA
		Rseq	GCTGCCTGCAGTATAATCCCT
VPS35-HiBiT	ENST00000299138.12	gRNA	GAAGGTCTCATCCTTTAAAA
		ssODN	TTGCGCTTGCGGCGGGAATCACCAGAATCCGAGGGGCCAATTTATGAAGGTCTCATCCTTGTGAGCGGCTGGCGGCTGTTCAAGAAGATTAGCTAAAAAGGAAATAGCTCACCATACTCCTTTCCATGTACATCCAGTGAGGGTTTTATTACGCT
		Ftaq	CGGGAATCACCAGAATCCGA
		Rtaq	AGGCACAATCTATGGAAGGG
		wt-HEX	T+C+T+CAT+C+C+TTTAAA
		HiBiT-FAM	CAA+G+A+A+GATTA+G+CT
		Fseq	GAGTGTAACAGAAGCTCCTCA
		Rseq	CAGTCATGCTACTTGGGGTGA
GBA-HiBiT	ENST00000327247.9	gRNA	ACCTGTGGCGTCGCCAGTGA
		ssODN	GGCTTCCTGGAGACAATCTCACCTGGCTACTCCATTCACACCTACCTGTGGCGTCGCCAGGTGAGCGGCTGGCGGCTGTTCAAGAAGATTAGCTGATGGAGCAGATACTCAAGGAGGCACTGGGCTCAGCCTGGGCATTAAAGGGACAGAGTCAGC
		Ftaq	ATTCACACCTACCTGTGGC
		Rtaq	CCTGCTGTGCCCTCTTTAG
		wt-HEX	CG+C+CAG+T+GATG
		HiBiT-FAM	AT+TA+G+C+T+GAT+GGAG
		Fseq	CTAAACCGGTGAGGGCAATG
		Rseq	GGGGAAAGTGAGTCACCCAAA
VPS13C-HiBiT	ENST00000644861.2	gRNA	CCTGAGGTCTGTGATTAAGA
		ssODN	CAGCAGCAAAAATTGATGAAGCAGTCATCAGTGAGACTTCTCAGACCCCAATTGCCATCTGTGAGCGGCTGGCGGCTGTTCAAGAAGATTAGCTAATCACAGACCTCAGGGGCTCCAACAGGGAGAAAAAACAATCACTGGTCTTGTCTAT
		Ftaq	AGTGAGACTTCTCAGACCCC
		Rtaq	AGCAAGATAAAGCAGAGTGACTT
		wt-HEX	TG+C+CA+T+C+T+TAATC
		HiBiT-FAM	CGG+CTGG+CGG
		Fseq	TATGCTGTGGACTCAGTCGG
		Rseq	TCTGACCCATTTGGGTGGTG
HiBiT-PRKN	ENST00000366898.6	gRNA	ACCTACCCAGTGACCATGAT
		ssODN	GCGCATGGGCCTGTTCCTGGCCCGCAGCCGCCACCTACCCAGTGACCATGGTGAGCGGCTGGCGGCTGTTCAAGAAGATTAGCATAGGTACGTGGGTACCTGCCAGGTACAGCCTCTCTGCGCCGCCCCACGCC
		Ftaq	CATGGGCCTGTTCCTGG
		Rtaq	GTCATTGACAGTTGGCACC
		wt-HEX	A+C+CAT+G+A+TA+GGT
		HiBiT-FAM	CA+TG+GT+G+AG+CG
		Fseq	CGGTGACGTAAGATTGCTGG
		Rseq	GGCTTCGAACGCACACACT
PRKN-HiBiT	ENST00000366898.6	gRNA	ACTGGTTCGACGTGTAGCCA
		ssODN	GAACTGTGGCTGCGAGTGGAACCGCGTCTGCATGGGGGACCACTGGTTCGACGTGGTGAGCGGCTGGCGGCTGTTCAAGAAGATTAGCTAGCCAGGGCGGCCGGGCGCCCCATCGCCACATCCTGGGGGAGCAT
		Ftaq	GAACTGTGGCTGCGAGT
		Rtaq	AGAAAATGAAGGTAGACACTGGG
		wt-HEX	TC+G+A+C+GT+GTA+GC
		HiBiT-FAM	CG+GC+TGG+CGG
		Fseq	TCCCGACAAAAGTGACATGCT
		Rseq	TTTGTGTCATCCGGAGGCTG

Note: + = locked nucleic acid (LNA).

**Table 4 ijms-25-09493-t004:** List of probes and primer sets used for RT-qPCR.

Gene	Sequence	Reference	Vendor
Nanog	n/a	Hs02387400_g1	Applied Biosystems
OCT3/4	n/a	Hs04260367_gH	Applied Biosystems
Nestin	n/a	Hs04187831_g1	Applied Biosystems
FOXA2	n/a	Hs00232764_m1	Applied Biosystems
Nurr1 (NR4A2)	n/a	Hs01117527_g1	Applied Biosystems
GBA	n/a	Hs.PT.58.40746061	Applied Biosystems
GAPDH	n/a	Hs02786624_g1	Applied Biosystems
ACTB	n/a	Hs01060665_g1	Applied Biosystems
GBA	n/a	Hs.PT.58.40746061	IDT
ACTB	n/a	Hs.PT.39a.22214847	IDT
GAPDH	n/a	Hs.PT.39a.22214836	IDT
GBA-HiBiT-F	GACAATCTCACCTGGCTACTC	n/a	IDT
GBA-HiBiT-R	CTCCTTGAGTATCTGCTCCATC	n/a	IDT
GBA-HiBiT probe	TGGCGGCTGTTCAAGAAGATTAGCT	n/a	IDT
LgBiT-F	CCGATCAGATGGCGCAAATA	n/a	IDT
LgBiT-R	GGTTCCGTAGGGCAGAATTAC	n/a	IDT

**Table 5 ijms-25-09493-t005:** List of antibodies.

Cell type	Antibody	Catalog	Dilution
iPSC	Nanog	Abcam Ab21624	1:200
	TRA1-60	STEMCELL Technologies #60064	1:200 IF
	OCT3/4	Santa Cruz SC-8628	1:2000 IF
	SSEA-4	Santa Cruz SC-21704	1:500 IF
dNPC	Nestin	Abcam ab92391	1:500 IF
	SOX1	R&D AF3369	1:500 IF
	FOXA2	R&D AF2400	1:500 IF
	LgBiT	Promega N7100	1:100 IF 1:1000 WB
	HiBiT	Promega N7200	1:100 IF 1:1000 WB

**Table 6 ijms-25-09493-t006:** List of primers used for off-target analysis.

CLYBL-17-F	ACTTACCAGCAACCTCGGTG
CLYBL-17-R	CATAGCCCCAGTCTGTGCAA
CLYBL-6-F	TGTGTCAGGGCTAAGAGTGC
CLYBL-6-R	TGACAGGCACATGGGGTTAG
CLYBL-10-F	GGAATCGTTTGCAGCCAGTG
CLYBL-10-R	CTACCCATCGCTCGGTCAC
GBA-20-F	TGACCTCTGGCTTCCATCAAG
GBA-20-R	GACCTATCTCTCAGAGCTGCC
GBA-1-F	GAAGCAGATGACACCTTGGC
GBA-1-R	TAGTTGGATGAAGGCTCTGGC
GBA-5-F	GAGCTGACGCTATTCGGTTTG
GBA-5-R	TTGGATGTCTGCCCCTTCG

## Data Availability

The raw data supporting the conclusions of this article will be made available by the authors on request. The iPSC clones used in this study are available to the scientific community via the Neuro’s C-BIG repository.

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
