# Peer review of "An Inducible Luminescent System to Explore Parkinson’s Disease-Associated Genes"

_ijms, 2024, doi:10.3390/ijms25179493_

Round 1

Reviewer 1 Report

Comments and Suggestions for Authors

This manuscript by Gandy et al presents a study wherein the authors have developed an iPSC cell line with dox-inducible LgBiT followed by editing that line to contain (endogenous) HiBiT-tagged Parkinson’s disease (PD) proteins of interest. The general idea of making such a resource is applauded, but the execution, validation, and utility of the lines needs to be improved. In many instances, the data provided are difficult to interpret or even visualize. The way that data are presented are confusing and not intuitive. For example, RQ ‘delta Ct’ is shown. I think that the authors just mean RQ, which is the fold change derived from the delta delta Ct method. But how are these data normalized initially? To the housekeeping genes like GAPDH and ACTB (primers listed) then to what? Why do many of the graphs have seemingly no bars and no error?  Western blots have virtually no signal and need more controls and information to be useful. While many HiBiT-tagged lines have been generated, some are not validated aside from genomic insertion, so it is not clear whether these are actually useful to the scientific community, or if they should be redesigned. Overall, there are many opportunities to improve on this manuscript so that it is an excellent resource for the PD community.

Major comments:

1.        siRNA should be used to verify that all the HiBiT signal is coming from indicated proteins of interest, not just off-target checking. If you are HiBiT tagging what you think you are, then siRNA will eliminate the signal.

2.        Fig. 1H – is the y axis supposed to say 101, 102, or 10^1, 10^2, etc? How do these values compare to what would normally be obtained using the Lytic HiBiT Assay kit where LgBiT is supplemented in trans?

3.        Fig. 2H – did iLgBiT not amplify at all in the – dox?  This highlights the confusing nature of how the data are presented. Also, why is there essentially no inducibility in the dNPCs in this figure, but robust induction in Fig. 4H?

4.        Fig. 3 – please show some editing efficiencies for the heterogenous and clonal lines. How many clones were screened to obtain each of these lines? Why are the agarose gels used for band cutting not presented?

5.        Fig. 3E – why would all of the HiBit lines show the same luminescence? These are all under different endogenous promoters. So this potentially indicates that LgBiT is limiting in this system. Please supplement LgBiT and show that the signal does not change to show that LgBiT produced in the cells is not limiting. Otherwise the signal in these cells may not truly reflect the levels of the endogenous HiBiT tagged proteins (i.e. a ceiling may have been reached, and you don’t know how much higher the true signal is above that ceiling).

6.        Fig. 4H and J – don’t agree with respect to iLgBiT

7.        What is being used for HiBiT detection via western? Is this a HiBiT antibody? Or HiBiT blotting?  Please indicate approximate molecular weights of the presented bands.

8.        How are Fig. 3E and 5B performed?  They cannot be performed as described in the methods since this does not involve using control HiBiT-tagged proteins. Were cells lysed and then only substrate added?  LgBiT only came from overproduction in the cell?

9.        Why was no live cell assay used?  This is really the main advantage of incorporating LgBiT into the cell line itself.

Minor

1.        All the tables included in the beginning of the manuscript that are not data are better suited to go to supplemental information.

2.        Fig. 1A, B, C – difficult to read some of the text in the printed copy, please fix

3.        What do the ‘+’ signs mean in the sequences in Table 2?

Reviewer 2 Report

Comments and Suggestions for Authors

Gandy et al. proposed using HiBiT-LgBiT to target these dysfunctional proteins associated with Parkinson’s disease. The authors have employed CRISPR-Cas9 gene editing to generate isogenic pairs of induced pluripotent stem cells (iPSCs), a robust method for ensuring genetic consistency across experimental conditions. To target PD, the authors focus on genes implicated in PD, such as GBA1, LRRK2, and SNCA, among others. The functional validation of these genes using the HiBiT-LgBiT system is directly relevant to the molecular mechanisms underlying PD. Moreover, the inducible luminescent-based assay has the potential for high-throughput screening of compounds that could modulate the activity of PD-associated proteins, which is valuable for drug discovery.

Overal, the work demonstrates that the HiBiT-LgBiT technology is a promising tool for advancing the understanding of PD's molecular mechanisms. The methodology was logically sound, and sufficient experiment details were provided. It could be accepted for publication after addressing the following points.

1. The font size and clarity of labels in the figures require improvement for better readability. The authors should ensure that all textual elements are of adequate size and are sharp to avoid ambiguity.

2. The format of the Tables should be improved. In addition, some tables, such as Tables 2, 3, 5, could be relocated to the supplementary information.

3. The experimental details of Figures in the supplementary information should be provided.

4. The manuscript should include information on the reproducibility of the findings, including the number of biological replicates and the consistency of results across experiments.

5. The authors claimed that the Doxycycline-inducible luminescence system was a sensitive method for quantifying HiBiT-tagged PD-associated protein. To strengthen this claim, they should present empirical data demonstrating the system's sensitivity and efficiency, including relevant statistical analyses.
